



# Options to correct local turbulent flux measurements for large-scale fluxes using a LES-based approach

Matthias Mauder[1], Andreas Ibrom[2], Luise Wanner[1], Frederik De Roo[3], Peter Brugger[4], Ralf Kiese[1], Kim Pilegaard[2]

[1]Institute of Meteorology and Climate Research – Atmospheric Environmental Research, Karlsruhe Institute of Technology, 82467 Garmisch-Partenkirchen, Germany
[2]Dept. of Environmental Engineering, Technical University of Denmark (DTU), Kgs. Lyngby,2800, Denmark
[3]Development center for weather forecasting, Norwegian Meteorological Institute, 0313 Oslo, Norway
[4]Wind Engineering and Renewable Energy Laboratory (WiRE), École polytechnique fédérale de Lausanne (EPFL), CH-1015 Lausanne, Switzerland

*Correspondence to*: Matthias Mauder (matthias.mauder@kit.edu)

**Abstract.** The eddy-covariance method provides the most direct estimates for fluxes between ecosystems and the atmosphere. However, dispersive fluxes can occur in the presence of secondary circulations, which can inherently not be captured by such single-tower measurements. In this study, we present options to correct local flux measurements for such large-scale transport based on a non-local parametric model that has been developed from a set of idealized LES runs for three real-world sites. The test sites DK-Sor, DE-Fen, and DE-Gwg, represent typical conditions in the mid-latitudes with different measurement height, different terrain complexity and different landscape-scale heterogeneity. Different ways to determine the boundary-layer height, which is a necessary input variable for modelling the dispersive fluxes, are applied, either from operational radio-soundings and local in-situ measurements for the flat site or from backscatter-intensity profile obtained from collocated ceilometers for the two sites in complex terrain. The adjusted total fluxes are evaluated by assessing the improvement in energy balance closure and by comparing the resulting latent heat fluxes with evapotranspiration rates from nearby lysimeters. The results show that not only the accuracy of the flux estimates is improved but also the precision, which is indicated by RMSE values that are reduced by approximately 50%. Nevertheless, it needs to be clear that this method is intended to correct for a bias in eddy-covariance measurements due to the presence of large-scale dispersive fluxes. Other reasons potentially causing a systematic under- or overestimation, such as low-pass filtering effects and missing storage terms, still need to be considered and minimized as much as possible. Moreover, additional transport induced by surface heterogeneities is not considered.

## 1 Introduction

Eddy-covariance (EC) measurements provide fundamental data for the development of numerical models in meteorology, hydrology and biogeosciences. In order to produce accurate flux estimates, a series of physically-based corrections are usually applied (Aubinet et al., 2012). Most of them are undisputed and are therefore used in standardized data processing strategies





(Aubinet et al., 2000; Mauder et al., 2013; Sabbatini et al., 2018). Nevertheless, researchers typically find a general systematic underestimation of the sum of the turbulent sensible and latent heat flux ($H + \lambda E$) by 10 to 30%, when these are validated against the available energy at the surface, i.e. the difference between net radiation and ground heat flux at the surface ($R_n - G_0$) (Hendricks-Franssen et al., 2009; Stoy et al., 2013; Wilson et al., 2002). Several studies indicate that the majority of this systematic bias is caused by dispersive fluxes, which arise from correlation of spatial variations of Reynolds mean variables and are influenced by the heterogeneity below the scale of the spatial averaging. Dispersive momentum fluxes are typically neglected when applying a volume averaging operator to describe the turbulent flow in plant canopies (Lee, 2018). However, these dispersive fluxes can be significant in the context of the surface for the total transport above the canopy, where they are a result of secondary circulations (Mauder et al., 2020). These circulations develop under convective conditions and are superimposed on the mean flow. They comprise distinct phenomena, which are large-scale turbulent organized structures over homogeneous surfaces and thermally induced mesoscale circulations over heterogeneous surfaces (Inagaki et al., 2006; Kanda et al., 2004; Steinfeld et al., 2007). Both phenomena contribute to the transport of momentum and scalars between the surface and the atmosphere, but can inherently not be captured by single-tower measurements (Etling and Brown, 1993).

A number of correction methods have been tested in the past in order to compensate for this systematic bias by distributing the surface energy balance (SEB) residual, either entirely or almost entirely towards the sensible heat flux (e.g. Charuchittipan et al., 2014; Ingwersen et al., 2011), or according to the Bowen ratio (e.g. Mauder et al., 2013; Twine et al., 2000), or entirely to the latent heat flux (e.g. Wohlfahrt et al., 2010). However, an evaluation of these different SEB closure adjustment options remained somewhat inconclusive as to which of the methods under investigation is preferable (Mauder et al., 2018). Now, a novel method is available that is, in contrast to the previous methods, physically-based and semi-empirical in character, meaning it relates the dispersive fluxes (which are causing a systematic bias in single-tower EC) to the non-dimensional, non-local scaling variables $u_*/w_*$ and $z/z_i$, where $u_*$ is the friction velocity, $w_*$ is the convective velocity scale, $z$ is the height above ground and $z_i$ is the boundary-layer height. The resulting functional relationships are fitted to a set of large-eddy simulations (LES), thereby representing a deeper understanding of the underlying transport processes (De Roo et al., 2018). As a result, we obtain two different correction equations, one for $H$ and one for $\lambda E$:

$$H_{\text{disp}} = \frac{F_{1H}(u_*/w_*)F_{2H}(z/z_i)}{1-F_{1H}(u_*/w_*)F_{2H}(z/z_i)} H_m \, , \tag{1}$$

$$\lambda E_{\text{disp}} = \frac{F_{1E}(u_*/w_*)F_{2E}(z/z_i)}{1-F_{1E}(u_*/w_*)F_{2E}(z/z_i)} \lambda E_m \, , \tag{2}$$

where the index "disp" stands for dispersive, representing the dispersive flux contribution that needs to be added as a correction to the measured fluxes, as indicated by the index "m". Please note that this correction is only applicable to unstable stratification, i.e. when the Obukhov length $L < 0$, because secondary circulations and the associated dispersive fluxes are restricted to these conditions. $F_{1H}$, $F_{2H}$, $F_{1E}$, and $F_{2E}$ are semi-empirical functions:

$$F_{1H} = 0.197 \exp(-17.0 \; u_*/w_*) + \; 0.156 \tag{3}$$

$$F_{1E} = 0.224 \exp(-14.0 \; u_*/w_*) + \; 0.071 \tag{4}$$





$$F_{2H} = 0.21 + 10.69 \; z/z_i \tag{5}$$

$$F_{2E} = 0.27 + 9.99 \; z/z_i \tag{6}$$

These constants were derived by De Roo et al. (2018) as the results of a curve-fitting to their model output. Due to the limited

grid resolution of the LES, which employed a grid spacing of 5 m in the horizontal and 2 m in the vertical direction, these

functions cannot be expected to hold for measurement heights by $z_m$ below 20 m. In this case, De Roo et al. (2018) suggest

that the correction is scaled by the daily energy balance ratio $EBR_d$, analogously to the method of Mauder et al. (2013).

Nevertheless, the partitioning of the residual is based on the LES parameter study of De Roo et al. (2018):

$$H_{tot} = H_m + \frac{H_{disp}}{(H_{disp} + \lambda E_{disp})} \; Res \; , \tag{7}$$

$$\lambda E_{tot} = \lambda E_m + \frac{\lambda E_{disp}}{(H_{disp} + \lambda E_{disp})} \; Res \; ,$$

$$\tag{8}$$

$$Res = \; (H_m + \lambda E_m)\left(\frac{1}{EBR_d} - 1\right), \tag{9}$$

where the index "tot" stands for the total corrected heat flux and the variable *Res* stands for the SEB residual based on

independent field measurements, assuming that dispersion is the only significant cause for the SEB imbalance.

In this study, we will present a first real-world application of this new SEB closure correction method. More specifically, we
will apply the method to data from three different EC sites with different site characteristics, such as canopy height, surface
heterogeneity and terrain complexity. One of these sites is a tall forest with an aerodynamic measurement height of 23 m, so
that the absolute magnitude of the correction can be evaluated by comparing the overall SEB closure before and after the
correction. For two other sites, we will compare the resulting estimates for the latent heat flux with independently measured
lysimetric evapotranspiration (ET) measurements. This will allow us to address the following two research questions:

1.      How realistic is the partitioning approach of the SEB residual into latent and sensible heat flux fractions?

2.      How well can the absolute magnitude of the correction be estimated?

We will now present further details about these three test sites, including their instrumentation and data processing chain in

section 2, followed by the results in section 3. Then, we will discuss the implications of our findings including the possibility

to use this method for other sites in section 4 before we summarize our conclusions in section 5.



## 2 Methodology

### 2.1 Soroe beech forest site (DK-Sor)

The Soroe beech forest site (DF-Sor) is located in the central part of the Danish island of Zealand (55.4858694° N, 11.6446444° E, 40 m AMSL). It is surrounded by flat but heterogeneous terrain, which is characterized by a land-cover mix comprising forests, agricultural area and small settlements (Figure 1). The beech forest around the EC tower is called "Lille Boegeskov", which extends approximately 2.4 km in N-S direction and approximately 1.0 km in E-W direction. The forest itself is also heterogeneous, as some smaller patches inside this beech forest are covered with other species, mostly plantations

of coniferous trees. Another larger beech forest, which is located to the Northeast of the EC tower (Figure 1), may also contribute to the flux footprint at times, depending on wind direction and atmospheric stability.

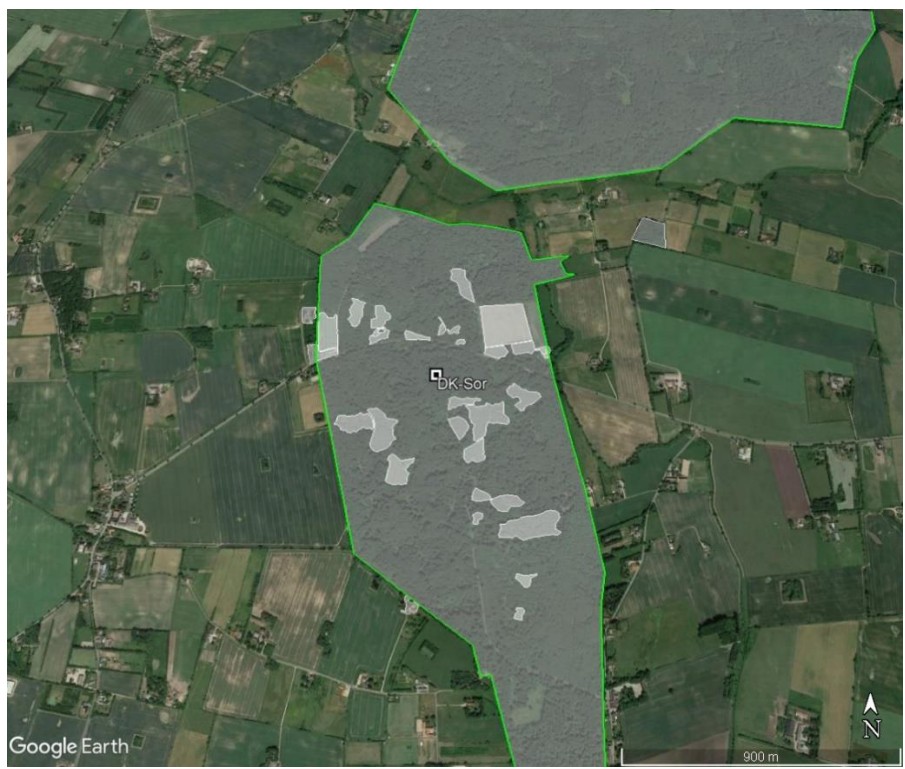

**Figure 1: Aerial image of the DK-Sor site. Beech forest is indicated by a polygon with a green edge and the exclusion areas, i.e. patches of coniferous trees, inside the Lille Boegeskov beech forest are indicated by polygons with a white edge, © Google Earth.**


### 2.1.1 Micrometeorological measurements

The EC-System is still very similar to the one developed in 1993 and operating since then at this site. A main feature is the long sampling tube that allows keeping the IRGA (LI-7000, Li-Cor, Lincoln, Nebraska, USA) in a temperature-controlled hut





close to the base of the 45 m tower. Since 2013, a Gill HS50 sonic anemometer is used at $z_m = 43.6$ m. The net-radiometer is
a combination of a pyrgeometer (CG4, Kipp & Zonen, Delft, The Netherlands) and a pyranometer (CM11, Kipp & Zonen)
each pointing both upwards and downwards. Contrary to the new ICOS set-up the devices are not ventilated. For details, see
Tab. 1 in Pilegaard and Ibrom (2020). Ground heat flux was observed with two self-calibrating heat flux plates (HFP01SC,
Thermal Sensors BV, Delft, The Netherlands). The eddy covariance raw data were processed as described in Pilegaard and
Ibrom (2020), applying the humidity dependent spectral dampening correction following Ibrom et al, (2007), but with a co-
spectral integration method of the total transfer function based on co-spectral models from Horst (1997) to calculate the flux
correction factor. The raw data were first processed with the software rcpm, developed by A. Ibrom, resulting in covariances
and variances. Further flux corrections and quality control were applied by using custom made R-scripts. The data set used for
this study covers period from 1 April until 31 December 2018.

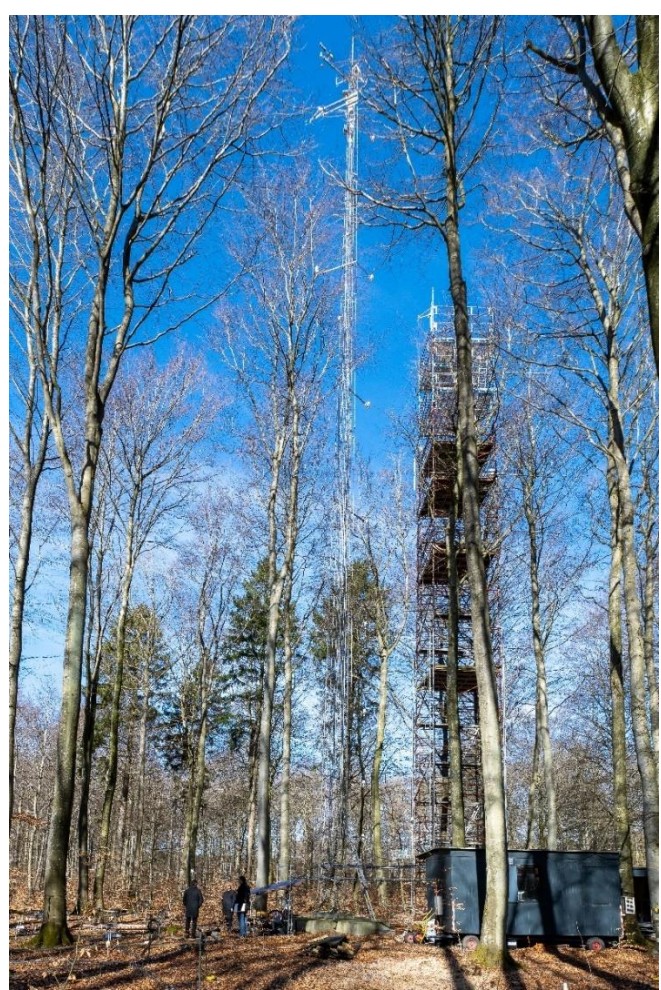


**Figure 2: Photograph of the 45 m mast and the nearby scaffolding tower at the DK-Sor beech forest site (photograph by Kim Pilegaard).**





### 2.1.2 Determination of the boundary-layer height

For the determination of the boundary-layer height at Soroe, we applied the method of Batchvarova and Gryning (1990), which
relies on friction velocity $u_*$, sensible heat flux $H$, and air density measured at the DK-Sor station. In addition, radio sounding
data for determining the morning temperature gradient in the free atmosphere were obtained from the worldwide repository
hosted at the University of Wyoming ([http://weather.uwyo.edu/upperair/sounding.html](http://weather.uwyo.edu/upperair/sounding.html)) for the station Schleswig (station
number 10035), which is located about 170 km west of this site. Despite the distance, it is the closest permanent radio sounding
station available, and it has a similar topographic setting as Soroe, so it is reasonable to assume the temperature gradient above
the boundary layer being similar. Based on these data, we calculated the potential temperature difference between the heights
of 1500 m and 500 m for the 6 UTC sounding on all days of the observation period.

### 2.1.3 Determination of the flux footprint

The flux footprint for the DK-Sor site was calculated for every 30-min averaging interval for the entire observation period by
using the simple two-dimensional parametric model (FFP) by Kljun et al. (2015). Besides the measurement height $z_m$, this
model requires 30-min data for the horizontal wind speed at the measurement height $u(z_m)$, friction velocity $u_*$, the Obukhov
length $L$, and the standard deviation of the cross-wind component $\sigma_v$ as input variables, which were readily available from the
EC system. In addition, the model requires $z_0$, which was set to a fixed value of 1.80 m, and $z_i$ for every 30 min interval, which
was determined according to the method of Batchvarova and Gryning (1990) as explained in section 2.1.2. The resulting flux
contributions from beech forest to the footprint of the DK-Sor EC dataset are presented in Figure 3 in form of a histogram.
The most common flux contribution class is from 0.7 to 0.75, meaning the 70% to 75% of the respective 30-min flux originates
from an area covered with beech forest based on a land cover map covering an area pf 4x4 km centred around the tower. The
median lies at a value of 0.704, meaning that in 50% of the 30-min intervals, more than 70.4% of the measured flux consists
of flux contributions from the beech forest. The average source contribution from beech forest is 75.6%.

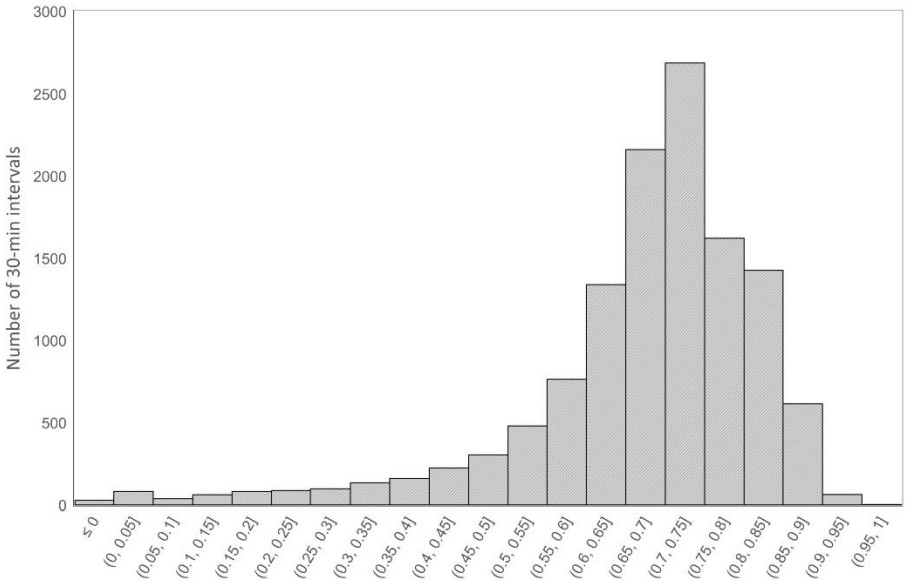

**Figure 3: Histogram of the flux footprint contributions of beech forest to the EC measurements at DK-Sor during the observation period from 1 April to 31 December 2018. Note that this dataset comprises all valid 30-min intervals, while the energy balance closure correction method is only applied for those with $z_i/L < 0$.**


The DK-Sor dataset comprised a total of 12469 30-min values. From those, 6572 values fulfilled the criterium of $z_i/L < 0$ of the De Roo et al. (2018) method and presence all four main components of the energy balance, so that the energy balance closure correction method could be applied. This dataset was then filtered using a threshold of 75% flux contributions from beech forest according to the quality requirements of Mauder et al. (2013). The number of measured data was further reduced

from 6572 to 4834, i.e. by 26%, due to the footprint filtering.

**2.2 TERENO Pre-Alpine grassland sites (DE-Fen, DE-Gwg)**

The two sites DE-Fen (Fendt, 47.8329° N 11.0607° E, 595 m AMSL) and DE-Gwg (Graswang, 47.5708° N 11.0326° E, 864 m AMSL) are located at flat valley-bottoms in the TERENO (Terrestrial Environmental Observatories) Pre-Alpine

Observatory, S-Germany. The DE-Fen site is surrounded by mildly complex terrain with differences in altitude on the order of 100 m while the DE-Gwg site is located in an area with differences in altitude on the order of 1000 m. The valley-bottoms in this region are usually grasslands, partially either managed as pasture and or as meadow and the slopes are often covered with forests up to the timberline. Both sites were chosen in a way so that their fetch is homogeneous and flat in all directions within a radius of 200 m, so that most of the flux contributions can be assumed to originate from grassland, which is the target

land use type. This has also been demonstrated through footprint calculations by Soltani et al. (2018).



### 2.2.1 Micrometeorological measurements

The instrumentation of the EC-systems at DE-Fen (Figure 4) and DE-Gwg (Figure 5) is nearly identical, comprising a CSAT3 sonic anemometer (Campbell Sci. Inc, Logan, UT, USA) and a LI-7500 infrared gas analyzer (Li-Cor Inc, Lincoln, NE, USA) for measuring the sensible and latent heat flux. The measurement height is 3.5 m above ground level. Net radiation is measured

at a height of 2 m above ground level by a four-component net-radiometer (CNR-4, Kipp&Zonen BV, Delft, The Netherlands) and ground-heat flux at the soil surface is measured by a combination of three self-calibration heat flux plates (HFP01SC), three soil temperature profiles (T106, Campbell Sci. Inc. Logan, UT, USA) and three soil water content profiles (CS616, Campbell Sci. Inc., Logan, UT, USA) following the PlateCal method of Liebethal (2005), which is a combination of heat flux plate measurements and a calorimetric approach. Further details about the additional meteorological measurements at these

sites can be found in Kiese et al. (2018).

The data processing follows the strategy for quality and uncertainty assessment of long-term EC measurements of Mauder et al. (2013). More specifically, we applied the double rotation method to align the coordinate system into the mean streamlines (Kaimal and Finnigan, 1994). We corrected for humidity fluctuations in the sensible heat flux measurement according to Schotanus et al. (1983). We compensated for spectral losses according to the method of Moore (1986) and corrected the latent

heat flux for density fluctuations following Webb et al. (1980). EC data were screened for steady-state conditions and well-developed turbulence according to a modified version of the method of Foken and Wichura (1996). The measurement period for this study is one entire year from 1 January until 31 December 2014.



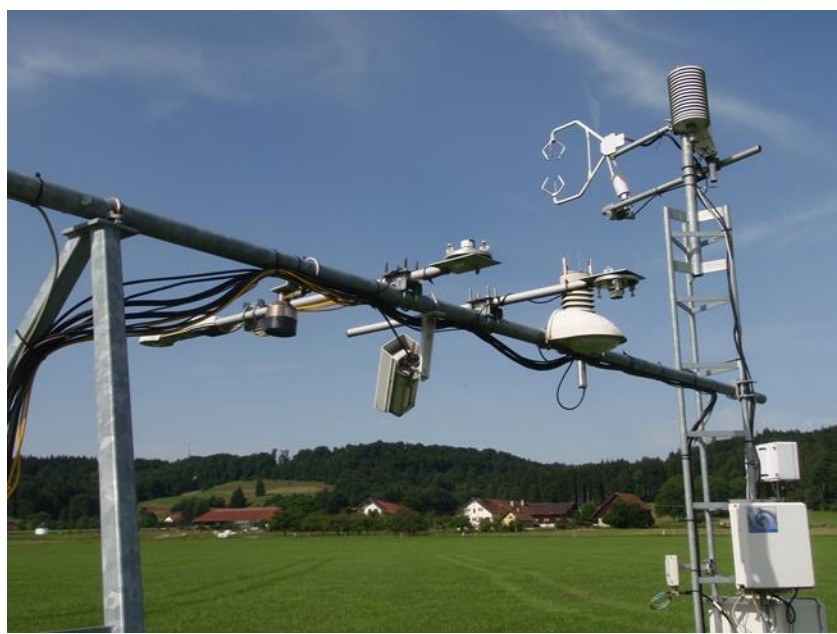


**Figure 4: Photograph of the eddy-covariance system at DE-Fen grassland site. The hill in the background has a height of about 100 m compared to the grassland in the foreground (photograph by Matthias Mauder).**

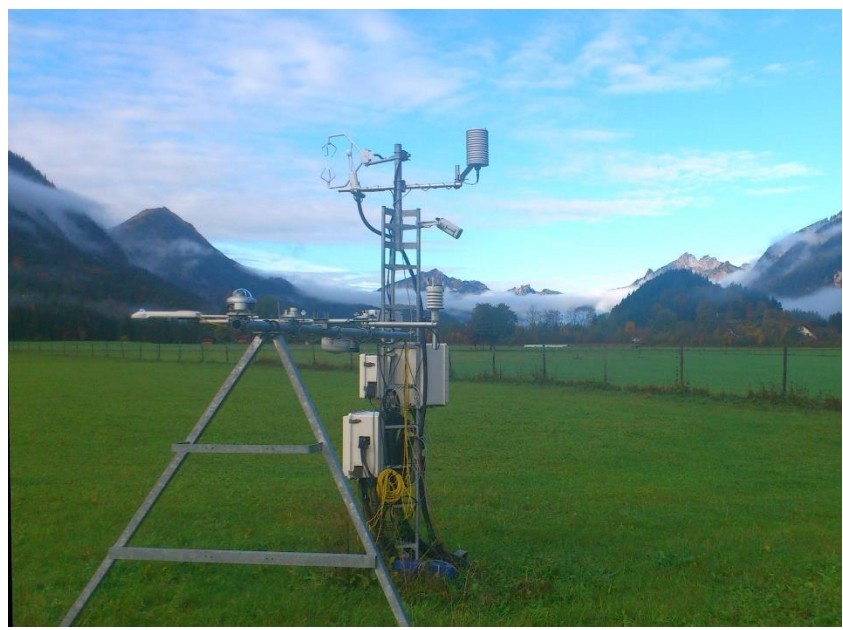

**Figure 5: Photograph of the eddy-covariance system at DE-Gwg grassland site. The mountains in the background are up to 1000 m**
**higher than the grassland in the foreground (photograph by Matthias Mauder).**

No $u_*$-filtering was applied because a decoupling of the canopy layer from the air above was not considered to be likely for these sites covered with short grass. However, the flux data are filtered using tests on well-developed turbulence and steady





state conditions (Foken et al., 2004; Foken and Wichura, 1996; Ruppert et al., 2006). In order to be able to compare the latent

heat fluxes measured at the TERENO Pre-Alpine grassland EC sites with the collocated lysimeters, daily sums of ET were

calculated. To this end, we applied the gap-filling approach of Reichstein (2005) based on a look-up table method by using the

REddyProc software (Wutzler et al., 2018) in the same way as Mauder et al. (2018) did this for an earlier comparison of SEB

adjustment methods.

### 2.2.2 Determination of the boundary-layer height

A ceilometer of type CL51 (Vaisala Oyi, Vantaa, Finland) was deployed at each of the two TERENO Pre-Alpine stations DE-

Gwg and DE-Fen. This instrument employs a pulsed diode laser LIDAR (Light Detection And Ranging) technology, where

short, powerful laser pulses are sent out in a vertical or near-vertical direction. The reflection of light (backscatter) caused by

aerosols, clouds, precipitation or another obscuration is analyzed. Backscatter profiles, which are averaged over 10 min, are

used to determine the boundary-layer height based on the maximum gradient method CL51 (Emeis et al., 2011; Münkel et al.,

2007) This method is based on the assumption of convective conditions, so that the aerosols are well-mixed throughout the

boundary-layer, while their concentration decreases sharply in the free atmosphere. The resulting values for $z_i$ were used as

input for the energy balance closure correction method of De Roo et al. (2018), which is only applicable for unstable conditions,

meaning that during those periods also the assumptions for the boundary-layer height retrieval method can be considered to

be fulfilled.

### 2.2.3 Lysimeter-measurements of evapotranspiration


As part of the TERENO SoilCan network the DE-Fen and DE-Gwg sites were equipped with fully automated lysimeter systems

which are operated with standardized sensor installations and measuring design (Pütz et al., 2016) in close vicinity (<500m)

to the respective eddy covariance systems (see 2.2.1). At both sites, evapotranspiration was measured from weighable

lysimeters (N=3) filled with intact soil cores (1 m², 1.4 m height), excavated at representative grassland locations in the

surrounding of the EC stations (Kiese et al., 2018). Water fluxes from at the bottom closed lysimeters are regulated by adjusting

matrix potential in 1.4m (TS1, Meter Group, Munich, Germany) inside the lysimeter to outside conditions measured in the

same depth in the surrounding soil. If the water tension in the lysimeter is higher than outside conditions, water is

pumped into a weighable tank via an underpressurized suction rake (SIC40, Meter Group, Munich, Germany) and vice

versa if the soil inside the lysimeter is drier than outside conditions. Grassland water fluxes of precipitation,

evapotranspiration, and groundwater recharge are derived from precision weighting of each lysimeter with three load cells

(Model 3510, Tedea-Huntleigh, Canoga Parl, CA, USA, precision of 10 g, equivalent to 0.01 mm water) and water tanks in 1-

min time intervals. Time series of lysimeter and water tank weights were quality checked before post-processing by applying

the adaptive window and adaptive threshold filter for separation of signal and noise (Fu et al., 2017; Peters et al., 2014). Daily





evapotranspiration rates in millimetres were calculated by summing up minute-based negative weight changes of the lysimeters
corrected by positive weight changes of water tanks representing lowering of lysimeter weights due to groundwater recharge.

# 3 Results

## 3.1 Case study not limited by $z_m$

First, we have a look at the dataset of DK-Sor (Figure 6), where the LES-based correction can be applied directly without the
need to adjust the correction factor with the measured $EBR_d$ because the aerodynamic measurement height $z_m$ is larger than
20 m there. The SEB closure is already relatively good on average in comparison with other sites with a slope of 0.94 and an
intercept of 3.26 W m$^{-2}$ of an orthogonal Deming regression (Manuilova et al., 2014). The scatter around this regression line
can be characterized by a Pearson correlation coefficient $r$ of 0.915. After application of the correction, the slope increases to
0.99 and the intercept stays almost the same with a value of 3.92 W m$^{-2}$. Also, the Pearson's $r$ is even slightly increased with
a value of 0.916. Overall, we can state that the SEB closure has clearly improved as a result of the correction with a regression
line very close to identity. As expected from De Roo et al. (2018), we found that the relative contribution of dispersive fluxes
to the total flux was larger for $H$ than for $\lambda E$, more specifically, H was increased by 6% on average and $\lambda E$ was increased by
4% on average. Please note that the correction is completely independent from measurements of the available energy at the
surface ($R_n - G$), the closeness of the regression line to the identity function is thus an empirical proof of the correction method.
However, no alternative measurements of either ET or $H$ are available for this beech forest site, so that we can only validate
the sum of the turbulent fluxes but not their partitioning between $H$ and $\lambda E$ here.

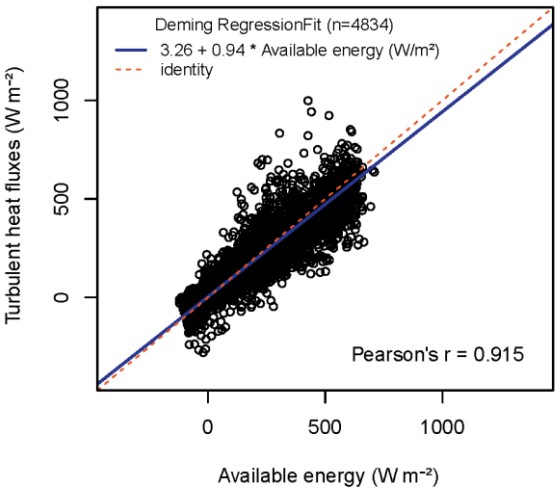
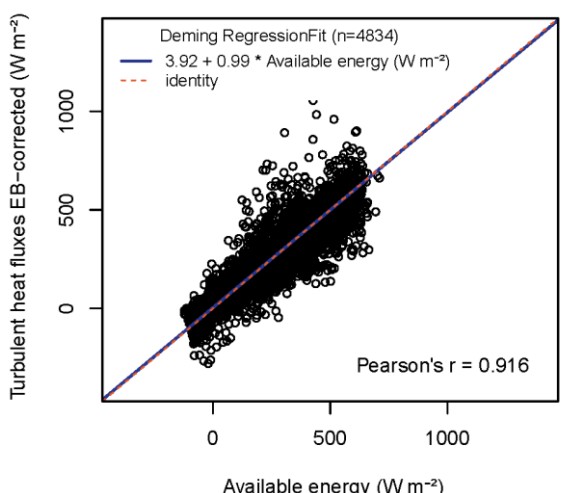

**Figure 6: Results of the orthogonal regression analysis of the sum of the turbulent heat fluxes ($H + \lambda E$) vs. the available energy ($R_n - G$) as measured (left panel) and after application of the EBC correction (right panel) for the DK-Sor dataset.**



In addition to this overall analysis of the energy balance closure, we also analyzed the seasonal variability of the mean diurnal cycle of the modelled dispersive fluxes in comparison with the measured residual (Figure 7). The agreement between both

curves is reasonable during daytime with generally positive values of up to 50 W m$^{-2}$. The modelled dispersive fluxes peak already before noon local time, which is 1100 UTC, while the measured residual peaks later in the early afternoon, at least for the months from April to September. During nighttime, the modelled dispersive fluxes are generally small and nearly zero. In contrast, the measured residual is often quite large and negative. This discrepancy reflects the fact that secondary circulations, and hence also the associated dispersive fluxes, are generally a phenomenon of the daytime convective boundary layer. At

night, other processes obviously contribute largely to the overall SEB residual, e.g. advection or storage terms, which are not considered in the model by De Roo et al. (2018). When comparing the different seasons with each other, we find that the daytime dispersive fluxes are smaller in the months from October to December than between April and September, which can probably be explained with less unstable conditions during this period of the year.

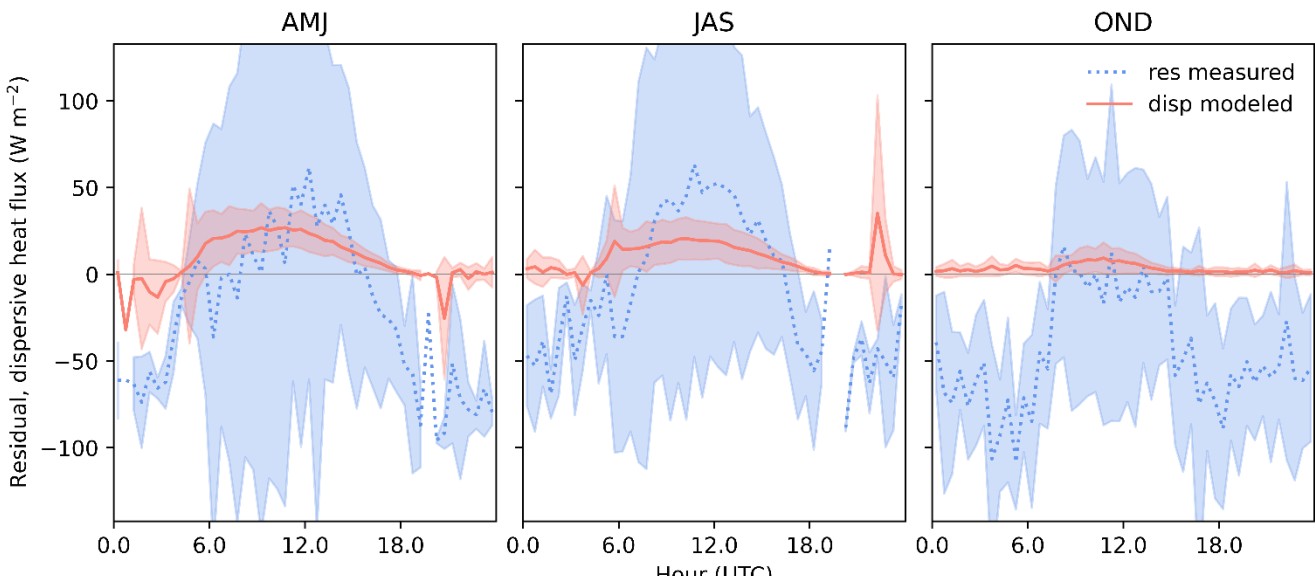

**Figure 7: Mean diurnal cycles of the measured residual (blue) and the modelled dispersive fluxes (red) for different seasons (AMJ: April, Mai, June; JAS: July, August, September; OND: October, November, December) for the DK-Sor dataset. The semi-transparent blue and red areas represent the respective standard deviations.**

### 3.2 Case study limited by $z_m$

Next, we apply the LES-based SEB correction to the grassland datasets of DE-Fen and DE-Gwg. At both sites, the effective measurement heights were less than 20 m, so that the correction had to be adjusted by the measured EBR$_d$ (eq. 7-9). As a result, the regression line after the correction is forced to be close to identity. And indeed, the regression slope is increased from 0.77 to 1.04 for the DE-Fen data (Figure 8) and from 0.70 to 1.02 for the DE-Gwg data (Figure 9). For both sites, also





the Pearsons's $r$ increases by 0.02 as a result of the correction, which is remarkable because it shows that this method also

reduces the random error and not only a systematic bias. It can be seen in the original data that the closure is better at low

energy fluxes, so that the data are slightly banana-shaped; this is nicely being corrected for.

Please note that the number of valid data points $n$ is reduced as a result of the correction by about 6-7% for both datasets, because we introduced two outlier criteria in order to avoid unrealistic fluxes when

   a)  $EBR_d < 0.5$ and $EBR_d > 1.5$, and

b)  $|(H_{res} + \lambda E_{res})| < 0.01$ W m$^{-2}$.

The reduction in sample size is also one reason why the resulting regression slopes are slightly larger than one and not identical to one as expected after applying this correction. Another reason is the slightly negative intercept, which is caused by a few data points with a negative sum of the turbulent heat fluxes. Note that no correction was applied to these data points by this method because is only necessary for unstable stratification.

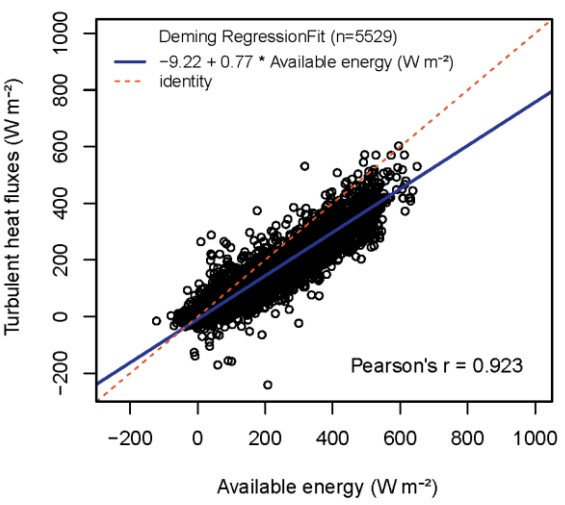
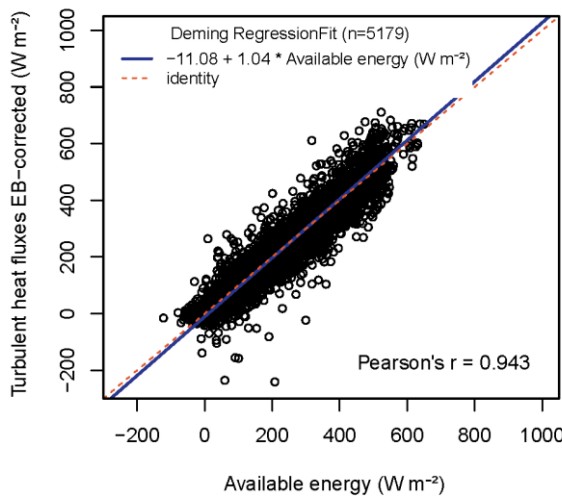

**Figure 8: Results the orthogonal regression analysis of the sum of the turbulent heat fluxes ($H + \lambda E$) vs. the available energy ($R_n - G$) as measured (left panel) and after application of the EBC correction (right panel) for the DE-Fen dataset.**



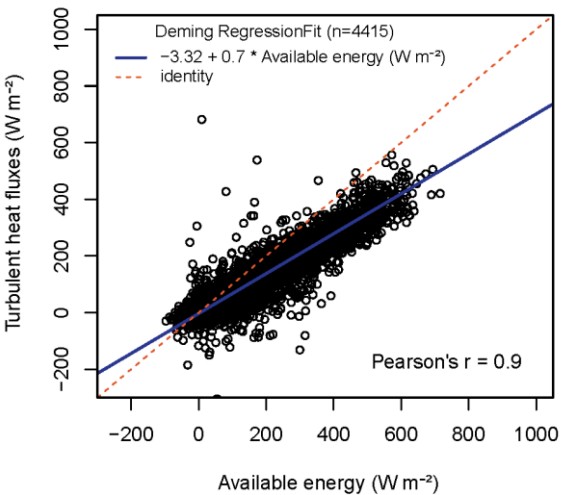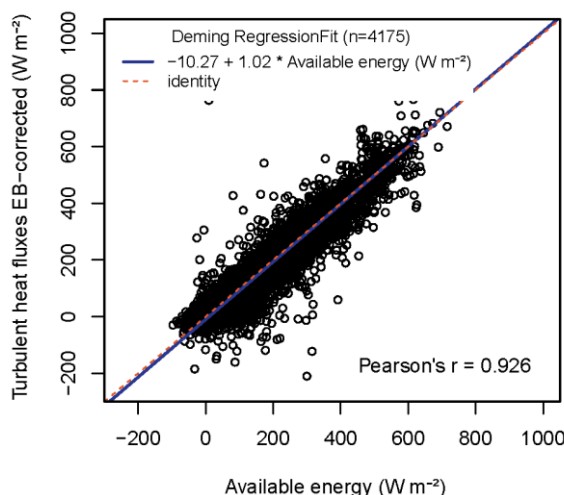

**Figure 9: Results the orthogonal regression analysis of the sum of the turbulent heat fluxes ($H + \lambda E$) vs. the available energy ($R_n - G$) as measured (left panel) and after application of the EBC correction (right panel) for the DE-Gwg dataset.**

### 3.3 Independent constraint on energy partitioning: water balance lysimeter

For the two grassland sites DE-Fen and DE-Gwg, the SEB closure correction needed to be scaled with the measured EBR due to the low measurement height. However, due to the nearby water balance lysimeters, we have the opportunity to test whether the partitioning of the SEB residual by this correction into the sensible and latent heat flux is realistic. To this end, we compared daily ET rates to four different options of correction for energy balance closure with these independent reference measurements; more specifically, these are

- No correction to the latent heat flux, i.e the entire residual is attributed to the sensible heat flux (Ingwersen et al., 2011)
- Bowen-ratio preserving partitioning of the SEB residual according to the daily $EBR_d$ (Mauder et al., 2013)
- Partitioning of the SEB residual according to the ratio between the sensible heat flux and the buoyancy flux forcing closure for every 30-min interval (Charuchittipan et al., 2014)
- LES-based correction for dispersive fluxes (De Roo et al., 2018)

For DE-Fen, we clearly see from Figure 10 that the first option without any correction produces systematically too low ET rates that are characterized by a regression slope of 0.70 with a moderate scatter indicated by a Pearson's r of 0.908. In contrast, the Bowen-ratio preserving method leads to an overestimation of ET and a regression slope of 1.09, while the scatter is slightly reduced with a Pearson's r of 0.915. The method of Charuchittipan et al. (2014) produces by far the largest scatter and the





lowest Pearson's r of 0.628, and the slope of 1.12 is even larger than that of the Bowen-ratio preserving method. Lastly, the LES-based method results in a slope of 0.9 and the highest Pearson's *r* of 0.937.

**Figure 10: Comparison of daily evapotranspiration (ET) estimates based on eddy-covariance measurements after different variants of energy balance closure correction with daily ET estimates based on lysimeter measurements for the DE-Fen dataset.**

Despite the large difference in terrain complexity between both grassland sites, the results for the DE-Gwg site are quite similar to those for DE-Fen (Figure 11). Again, we find the largest systematic underestimation of ET if no SEB closure correction is applied with a slope of 0.65. We also find the largest scatter and the lowest Pearson's *r* for the method of Charuchittipan et al.





(2014) with a value of 0.824 and the highest Pearson's *r* for the LES-based method with a value of 0.909. The Bowen-ratio

preserving method is slightly overestimating with a slope of 1.02 and an intercept of 0.33 mm d$^{-1}$, and the LES-based method

is slightly underestimating at least for larger ST values with a slope of 0.89 and an intercept of 0.23 mm d$^{-1}$.



**Figure 11: Comparison of daily evapotranspiration (ET) estimates based on eddy-covariance measurements after different variants of energy balance closure correction with daily ET estimates based on lysimeter measurements for the DE-Gwg dataset.**

Now, after this regression/correlation analysis, we will present the values for comparability and bias, which may provide

additional guidance to decide which of the four tested options leads to the best agreement with lysimetric ET estimates. These



results are shown in Table 1 and Table 2 for DE-Fen and DE-Gwg. For both datasets, we find a large negative bias of approximately –0.35 mm if no correction is applied, and a roughly equally large positive bias for the Bowen-ratio preserving method. The buoyancy-flux dependent method shows smaller biases than the latter two methods but results in a poorer

comparability of 1.40 and 0.855 mm for DE-Fen and DE-Gwg, respectively. Of all four methods, the LES-based correction leads to the best comparability with values of around 0.5 mm and it also leads to the lowest biases close to zero, i.e. well below 0.1 mm per day in absolute numbers.

**Table 1: Comparability/root-mean-square error (RMSE) of the different daily ET estimates in mm for the different SEB closure correction methods**

| RMSE (mm) | No correction (Ingwersen et al., 2011) | Bowen-ratio preserving (Mauder et al., 2013) | Buoyancy-flux dependent (Charuchittipan et al., 2014) | LES-based (De Roo et al., 2018) |
|---|---|---|---|---|
| **DE-Fen** | 0.811 | 0.799 | 1.40 | 0.551 |
| **DE-Gwg** | 0.858 | 0.755 | 0.856 | 0.561 |

**Table 2: Bias/mean error of the different daily ET estimates in mm for the different SEB closure correction methods**

| Bias (mm) | No correction (Ingwersen et al., 2011) | Bowen-ratio preserving (Mauder et al., 2013) | Buoyancy-flux dependent (Charuchittipan et al., 2014) | LES-based (De Roo et al., 2018) |
|---|---|---|---|---|
| **DE-Fen** | −0.365 | 0.413 | -0.064 | −0.010 |
| **DE-Gwg** | −0.346 | 0.355 | -0.184 | 0.041 |

**4 Discussion**

Closure of the SEB is to be expected for any given flux measurement site due to the first law of thermodynamics. A lack of closure indicates that not all assumptions of the EC method are sufficiently fulfilled in reality. Existing partitioning methods of the SEB residual are either based on no or weak physical basis. The newly proposed method of De Roo et al. (2018) is based on the understanding of the relevant transport process in the unstable boundary layer. It is well known from numerical and

observational boundary-layer studies that secondary circulations develop under typical daytime conditions, either cell-like or roll-like depending on the non-local stability parameter $z_i/L$ (Salesky et al., 2017). Both types of large-scale organized





structures fill almost the entire boundary layer and contribute to the overall vertical transport of scalars, such as temperature and humidity, by means of dispersive fluxes. In the presence of roll-like convection, the relative contribution is relatively small and constant over a wide stability range. As soon as cell-like convection develops, the relative contribution of dispersive fluxes

to the total flux increases sharply. Due to differences in typical vertical profiles of these scalars in the boundary layer, relative dispersive fluxes are larger for the transport of sensible heat than for the transport of latent heat in the surface layer (De Roo et al., 2018). No or very small dispersive fluxes are expected for near-neutral or stable stratification because secondary circulations do not develop under those conditions (Jayaraman and Brasseur, 2021). While plausible, this LES-based method has never been validated against real-world data before.

The generally good agreement between this model for dispersive fluxes and the independent reference measurements presented above is encouraging, but is the method of De Roo et al. (2018) really the solution to the longstanding energy balance closure problem? It certainly has the soundest physical basis of all the existing SEB correction approaches, since it is based on the theoretical process understanding that the underestimation of fluxes by single-tower EC systems during daytime is caused by dispersive fluxes that are generated by secondary circulations, and the semi-empirical correction model is the result of a

physically-based and systematic LES parameter study. Indeed, its application to the DK-Sor dataset leads to a nearly ideal overall SEB closure. In this case, the magnitude of the dispersive fluxes was modelled directly, and there was no need to use a scaling based on measurements of the available energy at the surface. Therefore, we were able to use these independent measurements of the available energy for validating the magnitude of the predicted SEB residual.

From the comparison of mean diurnal cycles between the modelled dispersive fluxes and the measured residual, we found that

the agreement is quite good during the day, meaning that the dispersive fluxes constitute indeed a major part of the missing flux, but during the night, other processes dominate, probably advection and storage terms (e.g. Moderow et al., 2009). Since sensible and latent fluxes are generally small at night, the overall energy balance can still be improved considerably through the correction for dispersive fluxes although these other terms contributing to the residual are not considered. During the summer months, strongly unstable conditions are more frequent and these are associated with cellular convection, which is

associated with large dispersive fluxes, while in fall and winter mildly unstable conditions are dominant, which lead to the formation of roll-like secondary circulations, which are associated with smaller dispersive fluxes.

The partitioning of the residual by the method of De Roo et al. (2018) is validated by the comparison of the resulting daily ET rates with independent lysimeter measurements for the DE-Fen and DE-Gwg datasets. The SEB closure after applying this correction is not quite as ideal as for the DK-Sor dataset but still much improved. One reason might be that the imbalance was

also initially less at DK-Sor and therefore the absolute correction is higher at the two grassland sites. Nevertheless, the resulting daily ET rates after applying the LES-based correction show the best agreement, i.e. the lowest bias and the lowest RMSE, of the four different methods under investigation when compared with the lysimeter data from both sites. The agreement is very good, despite the difference in scale and methodology between EC and lysimetry. This finding shows that the partitioning by the LES-based method is reasonable, and it is preferable to all other SEB closure adjustment methods that have been published

so far. In comparison to the other methods, the RMSE is approximately reduced by 50% through the LES-based method and



the bias becomes nearly zero. Hence, this new method is clearly a step forward towards more accurate flux estimates from EC systems which are of critical importance for improving meteorological and ecological models.

However, there are also two main disadvantages of the method by De Roo et al. (2018) that should be discussed. Firstly, this method requires the application of two outlier criteria in order to avoid unrealistic fluxes (see Sect. 3.2). These use somewhat

arbitrary and subjective thresholds and they lead to a reduction in data points by 6-7% for the two grassland datasets under study compared to the uncorrected data. However, this only applies to the cases that are limited by $z_m$. Theoretically, this $z_m$ limitation could be overcome by higher-resolution LES in the future, when this will be computationally feasible. If the correction method is not limited by $z_m$, as is the case for DK-Sor, no outlier criterium is needed, and hence, no reduction in valid data points can be listed. Secondly, this method was developed from the results of an LES that was driven by homogenous

lower boundary conditions and therefore does not include the effects of thermally heterogeneous surface heating on dispersive fluxes, which is relevant under certain realistic conditions as discussed e.g. by Zhou et al. (2019) and Margairaz et al. (2020). Nevertheless, the correction method shows a good comparison with the respective reference measurements for the three test sites, which are far from homogeneous on the landscape scale and also represent different levels of terrain complexity. However, for even more pronounced heterogeneities, especially when they are on a scale of roughly the boundary-layer height

or larger, we expect that this method is no longer valid (e.g. Eder et al., 2015).

The only test site DK-Sor, where the correction can applied directly, already has a relatively good SEB closure to begin with. This can be explained by the the rough surface in the surrounding in combination with the relatively high wind velocities that are typical for this region of Denmark. This leads to relatively high values of $u_*/w_*$, indicating forced convective conditions most of the time. In principle, an additional site with more unstable conditions would be interesting for this study as a

complement. However, such sites with high-quality energy-balance data, which also fulfil the criterion of $z_m > 20$ m are scarce. Theoretically, under more strongly unstable conditions, the LES-based correction would be much larger, and also the non-hydrostatic energy transfer might become more relevant (Sun et al., 2021), which otherwise is not the case.

We demonstrated different options how one can deal with the prerequisites for this method, which are an estimate of the boundary-layer height, and for cases that are limited by $z_m$, matching footprints between the measurement of the different

energy balance components and appropriate adaptation of spectral correction methods to the respective instrumentation. The latter two conditions are identical with the prerequisites of high-quality EC measurements in general and can therefore be assumed to be fulfilled, but in particular, the estimate of the boundary-layer height $z_i$ goes beyond the standard instrumentation of long-term EC sites, although it can also be helpful for other aspects related to long-term flux measurements (Helbig et al., 2020). We showed one option for how this important nonlocal scaling variable can be modelled from standard in-situ

measurements in combination with radio-sounding data that are freely available worldwide. For study areas that are located in mountainous regions, such as the TERENO Pre-Alpine sites DE-Fen and DE-Gwg, it is advantageous to use actual measurements of the $z_i$ by ceilometers.





## 5 Conclusions

For any operational application of such a method, it needs to be feasible, general and accurate, and our study addresses all of
these three aspects. Hence, we presented examples for the application of the novel LES-based SEB closure correction method
of De Roo et al. (2018) to three long-term EC sites of different land use and different canopy structure in the mid-latitudes.
With respect to the accuracy of the LES-based correction method, we found that it closes SEB almost perfectly on average for
the site that is not limited by $z_m$. For the other two sites, where the application of the correction method is limited by $z_m$, the
resulting bias is also close to zero when comparing the corrected latent heat fluxes with the ET estimate from nearby lysimeters.
Not only the accuracy of the flux estimates is improved by this method but also the precision, which is indicated by RMSE
values that are reduced by approximately 50%. Hence, our results demonstrate that this method has the potential to be applied
for operational application in long-term measurements for many sites around the world. Moreover, these results also suggest
that we can simulate the relevant transport processes in the unstable boundary layer realistically with the LES. The general
transferability of the idealized LES parameter study of De Roo et al. (2018) to the field has been successfully demonstrated.
However, this method is based on assumptions and has some remaining uncertainties. In its current form, it is limited to 30-
min block averages for the calculation of fluxes. Moreover, this flux correction method does not account for other sources of
bias or SEB non-closure than for the atmospheric transport through dispersive fluxes caused by secondary circulations, which
are restricted to unstable conditions. Therefore, it is important to note that storage terms should also be accounted for by an
adequate measurement set-up, if they are expected to be significant in magnitude, depending on the depth and the structure of
the canopy layer. In addition, great care should be taken in the site selection, the design of the measurement set-up, calibration
of the instruments, the implementation of all needed flux corrections and an effective set of quality control procedures (Mauder
et al., 2021), because this SEB closure correction cannot account for any of these aspects.

The promising results of this study will hopefully encourage further validation of this LES-based method for other sites around
the world, which perhaps even allow for a combined testing of magnitude and partitioning of the correction. It also remains to
be evaluated at what level of surface heterogeneity this method starts to lead to unrealistic results. We expect such a failure of
this method that was derived from completely homogeneous LES runs to occur when the formation of secondary circulations
is dominated by surface heterogeneity rather than self-organization of turbulence. In such cases, a set of LES runs that are
representative for a specific heterogeneous measurement site could potentially be used to overcome this limitation. Further
investigations are also warranted to establish whether a similar SEB closure correction could also be applicable for other trace
gases and scalars, e.g. $CO_2$.

## Acknowledgements

We thank DTU and KIT for supporting a three-month research stay of MM at DTU, which helped to develop the concept for
this study. LW's contribution was funded by the DFG as part of the CHEESEHEAD project (Grant number MA 6379/1-1)





The measurements were funded by ICOS Denmark. Funding for TERENO and ICOS-D was provided the Helmholtz Association and by BMBF. The support by the landowners of the TERENO sites and technical staff of KIT/IMK-IFU is appreciated. This work was partially conducted within the Helmholtz Young Investigator Group "Capturing all relevant scales of biosphere-atmosphere exchange - the enigmatic energy balance closure problem," which was funded by the Helmholtz-Association through the President's Initiative and Networking Fund and by KIT. Lioba Martin's assistance for the development of the R-script used to apply the energy balance closure correction is appreciated. We are also grateful to Sven-Erik Gryning for advice on how to determine boundary-layer height for the Soroe site.

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
