# Peer review of "Options to correct local turbulent flux measurements for large-scale fluxes using a LES-based approach"

_Atmospheric Measurement Techniques, 2021_

## Author Comment (AC1)

**Final response to the following Referee comment**

https://doi.org/10.5194/amt-2021-126-RC1, 2021

Anonymous Referee #2

Referee comment on "Options to correct local turbulent flux measurements for large-scale fluxes using a LES-based approach" by Matthias Mauder et al., Atmos. Meas. Tech. Discuss., https://doi.org/10.5194/amt-2021-126-RC1, 2021

The manuscript by Mauder et al evaluates a new method to correct turbulent flux measurements for the widely observed energy balance non-closure. The manuscript addresses an important research question and tests a new approach to solve a long existing problem for measurements of turbulent fluxes. I have a few comments that hopefully can improve the quality of this manuscript.

- Additional statistical analyses could be used to test if the flux corrections result in statistically significant improvements of the energy balance closure. The results qualitatively indicate improvements, but further statistical support of the findings would strengthen this study (e.g., through additional regression uncertainty analysis).

➔ We have chosen the orthogonal Deming regression method in order to account for the uncertainty of both, the x- and the y-variable. We have presented the Pearson's r-coefficient as a measure of goodness for the fits. In addition, we have conducted test on the significance between two correlations and included the results of these tests in the discussion section at the respective text passages.

- Unfortunately, the only site where the correction procedure can be applied directly has already a good energy balance closure, while the two other sites are characterised by a substantially worse closure. It would be helpful if sites with similar energy balance closures could be selected or at least if this issue would be discussed in more detail.

➔ We agree that the energy balance closure is already quite good for the DK-Sor site. We also agree that it would be good to do a similar validation for more sites with different energy balance closure. However, this is not possible within this study. Hence, we have followed the suggestions of the reviewer and extended the discussion in this direction:

The only test site DK-Sor, where the correction method can be applied directly, already has a relatively good SEB closure to begin with. The good closure can be explained by the rough surface in the surrounding in combination with the relatively high wind velocities that are typical for this region of Denmark. This leads to relatively high values of u_*/w_* , indicating forced convective conditions most of the time. In principle, an additional site with more unstable conditions would be interesting for this study as a complement. However, such sites with high-quality energy-balance data, which also fulfill the criterion of zm > 20 m are scarce. Theoretically, under more strongly unstable conditions, the LES-based correction would be much larger, and also the non-hydrostatic energy transfer might become more relevant (Sun et al., 2021). It is warranted that this correction method is further evaluated, particular for less windy sites with a sufficiently large aerodynamic measurement height, good fetch conditions and high-quality biometeorological measurements.

- Lastly, a new study by Sun et al. (https://doi.org/10.1029/2020JD034219) presenting a new hypothesis for the energy balance non-closure at flux tower sites related to nonhydrostatic energy transfer should be discussed in the manuscript. It would contribute

to a comprehensive discussion of the universality of the correction procedures outlined in the manuscript.

➔ We have included a reference to the paper by Sun et al. (2021) in the discussion section as part of the paragraph in response to the previous comment, see above.

---

## Author Comment (AC2)

**Final response to the following Referee Comment**

https://doi.org/10.5194/amt-2021-126-RC2, 2021

Anonymous Referee #3

Referee comment on "Options to correct local turbulent flux measurements for large-scale fluxes using a LES-based approach" by Matthias Mauder et al., Atmos. Meas. Tech. Discuss., https://doi.org/10.5194/amt-2021-126-RC2, 2021

The manuscript by Mauder et al., seeks to correct the local turbulent flux measurements based on a non-local parametric model from a set of LES. The authors used three different experimental sites. They showed that the accuracy of the turbulent flux estimates is improved after use the LES-based SEB closure correction method. These results are important for a better approximation of the closure of the SEB. The manuscript is well written. The text is clear and easy to read. Figures are very explanatory and aid understanding the manuscript. However, I have some minor concerns with the authors' methodology, which I feel should be resolved before publication.

1) As the LES simulations performed by De Roo et al. (2018) were an important part of the methodology used in this manuscript, I suggest that the authors create a subsection in the "Methodology" to inform readers the main steps in the work of De Roo et al. (2018) were used by the authors in this manuscript. If the authors do this, all the equations in the introduction would go into this subsection;

> ➔ We have followed this recommendation of Referee #3 and created a new subsection in the Methodology, which includes partially the equations from the introduction (section 2.1).

2) The experimental measurements performed in this study were carried out with very similar instruments/methodologies, for the three different experimentais sites investigated. Except for one of them: the height of the boundary layer. Sometimes the authores used a ceilometer data and sometimes they used the slab model (Batchvarova and Gryning (1990)) to identify the BLH. What is the impact on the results presented by the authors if one of these devices/methodologies does not adequately represent the height of the convective boundary layer?

> ➔ The sensitivity of the correction with respect to the value of the boundary-layer height $z_i$ can be directly seen from the correction equations F2H and F2E according to Eq. 5 and 6 of the revised version of this manuscript. Thus, the correction scales linearly with $z/z_i$, i.e. the ratio between the measurement height and the boundary layer height. In general, $z_i$ is typically much larger than $z$, and the correction amounts only to roughly 5-30% of the flux. As a consequence, a large absolute error in $z_i$ (e.g. 100 m) will only result in a relatively small error in $z/z_i$, and even a smaller relative error in the resulting flux. Therefore, the resulting flux is relatively robust to inaccuracies in $z_i$. Nevertheless, $z_i$ should be determined as accurately as possible. For the DK-Sor site, no continuous ground-based remote sensing data were available. Hence, we applied the method of Batchvarova and Gryning (1991), and the improvement in the energy balance closure, both by reducing the random and systematic deviations, shows that this method worked sufficiently well. This is not unexpected since

they have shown in their original paper for two different sites that the model is able to predict zi with an error on the order of 10%, and this has been confirmed in many other studies that have used this model for different sites around the world. Moreover, the DK-Sor site is located in relatively flat terrain at the scale of 100 km, where this model is expected to be applicable.

On the other hand, the DE-Fen and DE-Gwg sites are located in the Alpine foreland, which is highly complex terrain in comparison to the Danish site. For these sites, the Batchvarova and Gryning (1991) model is not applicable. Therefore, we used a method top determine zi from ground-based remote-sensing at these complex-terrain sites, which does not require the assumption of a flat homogeneous surface. The only main assumption of this method is well-mixed conditions in the boundary layer. Again, the improvement in the energy balance closure, particularly in reducing the random deviations, shows that the method works sufficiently well for this purpose. Similarly, to the method of Batchvarova and Gryning, the original paper by Münkel et al (2007) shows that the accuracy of this method is typically on the order of 10% or better in comparison to radio soundings.

We have added a paragraph along these lines at the end of the discussion section:

For a site with flat terrain on the landscape and regional scale, this important nonlocal scaling variable $z_i$ can be modelled from standard in-situ measurements in combination with radio-sounding data that are freely available worldwide. For study areas that are located in mountainous regions, such as the TERENO Pre-Alpine sites DE-Fen and DE-Gwg, it is advantageous to use continuous remote-sensing measurements of $z_i$ based on ceilometers. Both methods are expected to provide an accuracy on the order of 10% of $z_i$, which may lead to an error in the energy balance closure of the same magnitude, since it depends linearly on $z/z_i$. This correction only amounts to 5-30% of the fluxes. Hence, the resulting error of the flux is less than 5-30% of 10%, i.e. 0.5-3%. Moreover, the improvement in the energy balance closure, particularly in reducing the random deviations, shows that both methods work sufficiently well for this purpose.

Technical corrections;
L138: of 4x4 km;

➔ This typo has been corrected.

L200: ...(Emeis et al., 2011; Münkel et al., 2007). This method is...
➔ Thanks, we have added the full stop behind the references in brackets.

L211: 1.4 m
➔ We have added a blank space between the number and the unit.

L246-248: Isn't that logical? Hdis depends on H and the latter has higher values in summer
➔ We agree and have added that this can be explained by a combination of smaller sensible heat fluxes and less unstable conditions during the months October to December.